# Comparative Analysis of Clinical and CT Findings in Patients with SARS-CoV-2 Original Strain, Delta and Omicron Variants

**DOI:** 10.3390/biomedicines11030901

**Published:** 2023-03-14

**Authors:** Xiaoyu Han, Jingze Chen, Lu Chen, Xi Jia, Yanqing Fan, Yuting Zheng, Osamah Alwalid, Jie Liu, Yumin Li, Na Li, Jin Gu, Jiangtao Wang, Heshui Shi

**Affiliations:** 1Department of Radiology, Union Hospital, Tongji Medical College, Huazhong University of Science and Technology, Wuhan 430022, China; 2Hubei Province Key Laboratory of Molecular Imaging, Wuhan 430022, China; 3Department of Pharmacy, Wuhan Jinyintan Hospital, Wuhan 430022, China; 4Department of Radiology, Wuhan Jinyintan hospital, Wuhan 430022, China; 5Department of Diagnostic Imaging, Sidra Medicine, Doha 26999, Qatar; 6Xiangyang Central Hospital, Affiliated Hospital of Hubei University of Arts and Science, Xiangyang 441021, China

**Keywords:** SARS-CoV-2, Delta variant, Omicron variant, original strain, CT imaging

## Abstract

Objectives: To compare the clinical characteristics and chest CT findings of patients infected with Omicron and Delta variants and the original strain of COVID-19. Methods: A total of 503 patients infected with the original strain (245 cases), Delta variant (90 cases), and Omicron variant (168 cases) were retrospectively analyzed. The differences in clinical severity and chest CT findings were analyzed. We also compared the infection severity of patients with different vaccination statuses and quantified pneumonia by a deep-learning approach. Results: The rate of severe disease decreased significantly from the original strain to the Delta variant and Omicron variant (27% vs. 10% vs. 4.8%, *p* < 0.001). In the Omicron group, 44% (73/168) of CT scans were categorized as abnormal compared with 81% (73/90) in the Delta group and 96% (235/245, *p* < 0.05) in the original group. Trends of a gradual decrease in total CT score, lesion volume, and lesion CT value of AI evaluation were observed across the groups (*p* < 0.001 for all). Omicron patients who received the booster vaccine had less clinical severity (*p* = 0.015) and lower lung involvement rate than those without the booster vaccine (36% vs. 57%, *p* = 0.009). Conclusions: Compared with the original strain and Delta variant, the Omicron variant had less clinical severity and less lung injury on CT scans.

## 1. Introduction

Over three years after the first described COVID-19 patient of the original strain in December 2019 [1,2], multiple variants of concern of Severe Acute Respiratory Syndrome Coronavirus 2 (SARS-CoV-2) emerged, varying in transmissibility and severity [3]. The Delta variant was first identified in India in October 2020 and became the dominant strain detected globally in June 2021. Subsequently, the Omicron variant, which was first reported in South Africa on 24 November 2021, quickly replaced the Delta variant and became the predominant strain globally. The SARS-CoV-2 variants carry signature amino acid substitutions in key areas of the immunodominant spike protein, with evidence of altered virus characteristics [4]. Hence, the variances in clinical and imaging characteristics of SARS-CoV-2 variants and vaccine effectiveness gained public concern.

Numerous studies have revealed the clinical manifestations [5], imaging characteristics [6], and outcomes [7] of the first wave of SARS-CoV-2 (original strain). However, clinical and lung CT findings of the Omicron and Delta variants are lacking. Two recent studies [8,9] from the UK and South Korea have indicated that the CT severity of infection is lower for Omicron than for Delta. However, none had compared the original strain to the Omicron and Delta variants. In addition, previous research showed that (Artificial intelligence) AI algorithms could even achieve or exceed the performance of human experts in the detection and differential diagnosis of COVID-19 based on chest imaging [10,11,12]. However, the quantification of pneumonia infected with different SARS-CoV-2 variants by AI software was not addressed.

Despite clear evidence of fewer intensive care unit (ICU) admissions [13], the debate over vaccine effectiveness persists, especially against mutant strains. The Delta variant has mutations that make it highly infectious (more than 60% more infectious than the previous variants), slower to respond to the antibodies and therapy (reduced neutralization by antibodies from previous infections or vaccinations), and more likely to cause adverse outcomes [14]. Similarly, the evidence demonstrated that vaccine-induced immune protection might more likely be absent for Omicron compared to the prototypes and other SARS-CoV-2 variants [15]. The main essential element leading to the evolution of SARS-CoV-2 infection is the interaction with the host immune system. However, there is a need to understand how the new variants can lead to severe forms of the disease.

Timely understanding of the clinical characteristics and imaging manifestations of different SARS-CoV-2 variants was very crucial for clinical diagnosis and treatment. Therefore, this study aims to compare the clinical characteristics and clinical disease severity of patients infected with the Omicron and Delta variants and the original strain of COVID-19.

The rest of this paper was organized as follows. Section 2 showed how to include and exclude patients, the CT scanning method, and the statistical analysis used. We also compared the infection severity of patients with different vaccination statuses and quantified pneumonia by a deep-learning approach. In Section 3, the critical and statistically significant results are described. The significance of the results of this paper and the findings of this paper as they relate to previous research are discussed in Section 4. We also conclude the paper in Section 4.

## 2. Materials and Methods

### 2.1. Patients and Clinical Characteristics

Ethical approval for this study was obtained from the Ethics Committee of Wuhan Jinyintan Hospital (Approval Code: KY-2020-04-01, Approval Date: 1 April 2020). Informed consent was waived. Data were retrospectively collected on consecutive patients admitted to an infectious disease control hospital (Wuhan Jinyintan Hospital, a tertiary care center) at three different time points: SARS-CoV-2 Original strain from 1 February to 28 February 2020; Delta variant from 1 August to 31 August 2021; and Omicron variant from 7 December to 13 December 2022. Inclusion criteria were as follows: (a) polymerase chain reaction assay-proven SARS-CoV-2 and the variants of concern; and (b) >18 years old. Exclusion criteria included (a) patients who did not undergo CT scanning; and (b) inadequate CT quality. The standard for clinical severity (Appendix A Appendix A) and recovery was according to the ninth edition of “pneumonia diagnosis and treatment plan for new coronavirus infection” in China [16]. Ultimately, a total of 503 cases of SARS-CoV-2 infection were included in this study, including 245 cases of the original strain, 90 cases of the Delta variant, and 168 cases of the Omicron variant (Figure 1).

The medical records of each patient on admission were reviewed by one of four physicians (Y.F., N.L., X.H., Y.L.). Age, sex, underlying disease, the onset of symptoms, laboratory results on admission, and treatment measures were collected. The durations from the onset of disease to hospital admission and chest CT scan were recorded. Acute Respiratory Distress Syndrome (ARDS) was diagnosed according to the Berlin definition [17].

### 2.2. CT Imaging Acquisition

All CT scans were obtained with the patients in a supine position. CT was performed using one of the following CT scanners: SOMATOM Definition AS+, Siemens Healthineers, Forchheim, Germany. The scan ranged from the level of the upper thoracic inlet to the inferior level of the costophrenic angle. The CT parameters were as follows: detector collimation width, 64 × 0.6 mm, 128 × 0.6 mm; tube voltage, 120 kV. The tube current was regulated by an automatic exposure control system (CARE Dose 4D). No contrast agents were used.

Images were reconstructed with a slice thickness of 5 mm or 1.25 mm and an interval of 5 mm or 1.25 mm. In particular, 105/245 (42.8%) CT scans of patients with the original strain infected were of 5 mm slice thickness. The reconstructed images were transmitted to the workstation and picture archiving and communication systems (PACS) for multiplanar reconstruction (MPR) postprocessing.

### 2.3. Imaging Interpretation

The initial images and follow-up chest CT obtained from all patients were reviewed by two radiologists (HSS, a senior thoracic radiologist with 30 years of experience, and YQF, an attending radiologist with 19 years of experience in interpreting chest CT images). All Digital Imaging and Communications in Medicine (DICOM) images were analyzed from the CT studies without access to the clinical and laboratory data of the patients. The interpreters independently and freely assessed the CT features using both axial CT images and MPR images. After separate evaluations, disagreements, wherever found, were solved by discussion and consensus.

CT images of COVID-19 pneumonia were divided into the following four categories based on the Expert Consensus published by the radiological Society of North America (RSNA) during the first wave of SARS-CoV-2 [18]: typical appearance, indeterminate appearance, atypical appearance, COVID-19-negative Pneumonia (Appendix A). On each CT scan, the extent of lesion involvement was categorized as focal, multifocal, or diffuse. The predominant pattern was categorized as ground-glass opacity, consolidation, or reticular pattern. In addition, crazy paving, peribronchial wall thickening, pleural thickening, nodule or mass, tree-in-bud, pleural effusion, and halo sign, defined according to the Fleischner Society glossary [19] were also documented.

To quantify the extent of pulmonary abnormalities (all lesions, ground grass opacity (GGO), consolidation, and reticulation), a semiquantitative CT score [20] was assigned on the basis of the area involved in each of the five lung lobes: 0, no involvement; 1, <5%; 2, 5–25%; 3, 26–49%; 4, 50–75%; and 5, >75% involved area. The total CT severity score was calculated by summing the individual lobar scores (possible scores ranged from 0 to 25).

### 2.4. Quantitative CT Analysis by Artificial Intelligence

The quantitative CT analysis was performed by means of commercially available segmentation software (InferRead MCT Lung, Infervision, Europe GmbH Wiesbaden Germany), an AI solution specifically developed for the diagnosis and management support of COVID-19 pneumonia (Appendix A). Among its features, the algorithm module included automated segmentation of the core features of COVID-19 lung lesions and the segmentation of the lung lobes. The core algorithm is based on a deep convolutional neural network structure and uses the U-net network structure as the core segmentation network [21]. UNet (U-Net, https://lmb.informatik.uni-freiburg.de/people/ronneber/u-net/ (accessed on 12 March 2023)), as a semantic segmentation framework, can effectively process the features of CT images of different scales and generate segmentation results. UNet + Xception structure is proposed to be adopted in the model structure. UNet is a semantic segmentation framework that can effectively process the features of CT images of different scales and generate segmentation results. Xception is used as the backbone of UNet feature extraction. Xception can effectively avoid overfitting in the structure design. After the completion of training, the model can accurately segment the lung abnormalities area and calculate the percentage of pneumonia lesions in the overall lung volume. In a previous study, this AI algorithm achieved segmentation accuracy of a DICE coefficient of 0.8481 on internal test sets [22]. All thin-slice CT scans (1.25 mm) were imported into the software for calculation of the volume, CT value, and the proportion of pneumonia. Then, two radiologists (X.H. and LC, both with more than five years of experience in CT diagnosis) manually corrected the segmentation results for all patients.

### 2.5. Statistical Analysis

The analyses were performed using SPSS Statistics (SPSS, version 21, IBM, Chicago, IL, USA). Distribution normality was assessed using the Kolmogorov–Smirnov test. Normally, non-normally distributed data and categorical variables were expressed as the mean ± standard deviation, median (interquartile range), and frequency (percentage), respectively. Independent-sample Student’s *t*-test and one-way ANOVA were employed for comparing normally distributed variables between different groups, the proportions, and the means between the groups, respectively. Between-group differences in nonnormally distributed variables were compared with the Kruskal–Wallis or Mann–Whitney U tests. Binary or multiple logistics regression was adopted to correct confounding factors affecting different strain groups and different vaccination statuses, including age, gender, underlying diseases, and vaccination status. In the regression model, the cutoff value of the dependent variable was selected according to the median (IQR).

## 3. Results

### 3.1. Demographics and Clinical Characteristics on Admission

A total of 503 patients infected with the original strain (245 cases), Delta variant (90 cases), and Omicron variant (168 cases) were retrospectively analyzed (Table 1). Compared with the original strain, both the Delta variant and Omicron variant groups more commonly occurred in younger patients without underlying diseases. However, there was no significant difference in gender among the three groups (*p* = 0.211). The majority of patients with the original and Delta variants were moderate type (69% vs. 71%) for clinical severity, while more than half of Omicron infections were mild type (57%). Among the concomitant symptoms, fever (original, 88%, Delta, 64%, and Omicron, 75%) was the most common concomitant symptom in all three groups. However, the maximum temperature in original and Omicron patients was significantly higher than that in the Delta group (38 ± 1.6 °C vs. 38.2 ± 1.2 °C vs. 37 ± 1.0 °C). Notably, more than half of patients with the original strain 132 (54%) had asthma symptoms, while the Delta variant and Omicron variant were less likely to have asthma (5.6% and 6.5%). In addition, patients infected with the Delta variant were more likely to have anosmia and conjunctivitis than those infected with the original and Omicron variants (*p* < 0.05), and patients infected with the Omicron variant were more likely to have pharyngeal pain than the other two groups (31% vs. 23% vs. 6.6%, *p* < 0.001).

The disease severity has significantly decreased from the original strain to the Delta variant and Omicron variant (26.5% vs. 10% vs. 4.8%, *p* < 0.001). After controlling for gender, underlying diseases, and vaccination status with multiple logistics regression, SARS-CoV-2 strain type was significantly negatively correlated with clinical severity (OR = 0.306, *p* < 0.001) (adjusted OR = 0.306, *p* < 0.001, Appendix A). Among the three different SARS-CoV-2 strains, Delta had the longest duration of infection, and the longest hospital stays, whereas Omicron had the shortest (*p* < 0.05, Appendix A). Regarding treatment, the proportion of patients who required oxygen and steroid therapy was significantly higher in the Delta group than in Omicron (*p* < 0.05); no significant difference was found in any other treatment methods among different groups. Of the patients infected with the original strain, 17% (42/245) required admission to the intensive care unit (ICU), 26.5% (65/245) experienced acute respiratory distress syndrome (ARDS), and 10% (25/245) died in hospital. ICU occupancy rate (2.2% vs. 3.6%) and ARDS experience (1.1% vs. 0) were significantly decreased in the Delta group and Omicron group, and no deaths were reported.

### 3.2. Laboratory Findings

The median cycle threshold (Ct) values on the day of the first positive RT-PCR were 23.6 (Ct range 11.3–19.8) for the ORFa1b gene and 22.6 (Ct range 14.1–28.3) for the N gene of Delta groups (Appendix A) and the median cycle threshold (Ct) values on the day of first positive RT-PCR were 29.9 (Ct range 24.8–33.2) for ORFa1b gene and 27.6 (Ct range 24.6–32) for N gene of Omicron groups. The lymphocyte count in patients with the original strain was significantly lower than that of the two variants. Delta variant infection was associated with higher lactate dehydrogenase (LDH), D-dimer, interleukin-6 (IL-6), and C-reactive protein (CRP) compared with the Omicron variant, while no difference was found in the other laboratory findings.

### 3.3. Chest CT Score and Features

In the Omicron group, only 43.5% (73/168) of CT scans were categorized as abnormal compared with 66.7% (60/90) in the Delta group and 96% (235/245) in the original strain (*p* < 0.001, Table 2). A gradual decreasing trend was observed across the groups in total CT score (original 14 (IQR 9.0,20.0), Delta 6.0 (IQR 3.0,8.5), and Omicron 5.0 (IQR 3.0,10), *p* < 0.001, Table 3). In the binary logistics regression analysis, the total CT score > 5 was significantly negatively correlated with strain type (OR = 0.384, *p* < 0.001) (adjusted OR = 0.384, *p* < 0.001, Appendix A). Moreover, the frequency of indeterminate appearance (original 1.3% vs. Delta 11% vs. Omicron 18%, *p* < 0.001) and atypical CT appearance (original 1.7% vs. Delta 14% vs. Omicron 16%, *p* < 0.001) were gradually increased. The infection of the three strains mainly involved the lower lobes of both lungs (90% vs. 78% vs. 77%, *p* = 0.007) and presented subpleural distribution (96% vs. 85% vs. 78%, *p* < 0.001), and the lesion form was dominated by GGO (75% vs. 62% vs. 66%, *p* = 0.093). Patients infected with the Omicron variant mainly presented as multifocal or focal ground-glass opacities (GGOs) or consolidations, whereas original and Delta variants had significantly higher rates of diffuse lesions than Omicron variants (74% vs. 47% vs. 33%, *p* < 0.001, Figure 2). Patients infected with the Omicron variant had a higher prevalence of nodules, tree-in-bud, and halo signs than patients with the original strain or Delta variants infections (*p* < 0.001, Figure 2). As shown in Appendix A, the AI evaluation results of pneumonia lesions infected by different variants showed that the lesion volume, lesion CT value, and lesion proportion of the whole lung, left lung, and right lung of patients in the three groups were gradually reduced, which was consistent with the semiquantitative CT score results.

### 3.4. Patient Clinical Characteristics by Vaccination

As shown in Table 3, among the patients infected with the Delta variant, 22 (24%), had not received the COVID-19 vaccine (unvaccinated group), and 68 (76%) had received at least one dose of vaccine (vaccinated group). In patients with the Delta variant, the vaccinated cases had slightly less clinical severity and shorter in-hospital time (31 (IQR 24,41) days vs. 26 (IQR 19,33) days, *p* = 0.02) than unvaccinated cases (*p* < 0.05, Appendix A). However, no significant differences were found in the pulmonary involvement rate or CT scores or regarding vaccination status (*p* > 0.05, Table 3). Among Omicron variant patients, 36% (61/168) did not receive the third dose of the COVID-19 vaccine (the group that did not receive the third dose), and 64% (107/168) received the third dose of COVID-19 vaccine (the group that received the third dose). Patients with the Omicron variant who received the third dose had less clinical severity (*p* = 0.015, Appendix A) than those who did not receive the third dose. After controlling for age, gender, and underlying diseases with multiple logistics regression, the third dose acceptance was significantly negatively correlated with clinical severity (adjusted OR = 0.313, *p* = 0.009, Appendix A). Patients with the Omicron variant who received the third dose had a lower lung involvement rate (36% vs. 57%, *p* = 0.009) than those who did not receive the third dose. In multiple logistics regression, the third dose acceptance was significantly negatively correlated with lung involvement rate after adjustment for confounders (adjusted OR = 0.458, *p* = 0.02, Appendix A). However, no significant differences were found in the CT score or in-hospital time regarding vaccination status (*p* > 0.05, Table 3).

## 4. Discussion

In the present study, we found that the disease severity, extent of pneumonia, and typical appearance of pneumonia gradually decreased from the original strain to the Delta variant and Omicron variant group. Moreover, vaccinated patients had slightly less clinical severity and shorter in-hospital time than unvaccinated patients in the Delta group, while patients without a booster vaccination had more serious clinical severity and higher lung involvement rate than those with booster vaccination in the Omicron group. Since the CT features of variants may confuse radiologists and delay the diagnosis of COVID-19 in occasional circumstances, then we need to evaluate whether CT apparencies remained consistent or changed when new variants emerged.

Previous reports have indicated that the clinical severity of infection is lower for Omicron than for Delta [23,24,25]. The present study confirms these findings, as the disease severity has significantly decreased from the original strain to the Delta variant and Omicron variant. It may be related to weaker virulence and a younger, fully vaccinated population with no underlying disease. This, however, does not necessarily mean that the viral evolution leads to a lower severity. The risk of hospitalization appeared to increase when comparing Delta with original infections [26,27]. Another interesting finding was that throat pain, hyposmia, and conjunctivitis occurred more frequently in Delta strains than in the original strain and Omicron strains. A recent study [14] showed that the viral load of the Delta variant was significantly higher in patients with wild strain and that the virus enters the target cell in the nasopharynx through the angiotensin-converting enzyme ⅱ (ACE- ⅱ) [28]; hence, it was considered that the high incidence of pharyngeal discomfort in the mutant group was related to the high viral load in the nasopharynx, which also explains the longer drug loading and hospital stay in Delta patients in the current study. We also identified that Delta variant infection was associated with higher LDH, D-dimer, IL-6, and CPR compared with the Omicron variant, as these parameters were associated with more severe disease [29], and so this finding is consistent with the fact that Delta causes more severe illness than Omicron.

On the CT scan, there was a high proportion of negative CT (56%) for pneumonia in the Omicron group when compared to the original strain and Delta groups. This may be related to the fact that omicron viruses tend to invade the upper respiratory tract over the lower respiratory tract, which is also seen in other respiratory viruses, including other coronaviruses [30,31]. The frequency of typical CT appearance of COVID-19 pneumonia was significantly higher in original strain patients than in variant groups, which is similar to the results of a relevant recent publication [9]. The RSNA Consensus for Findings Related to COVID-19 on Chest CT was based on the first wave of SARS-CoV-2 (original strain) in 2020, which showed a diagnostic performance of a moderate sensitivity (68%) and high specificity (91%) in the real world [32]. On the other hand, patients infected with variant groups, especially Omicron, had a significantly higher prevalence of nodules, tree-in-bud appearance, and halo sign than in patients with the original strain. Maria et al. [8] found that bronchial wall thickening was more common in Omicron than in Delta infection. In our study, bronchial wall thickening was slightly more common in Omicron than in Delta infections (28% vs. 11.9%). Taken together, due to the high proportion of negative CT and the atypical appearances of pneumonia in Omicron, the limitations of using CT for COVID-19 diagnosis have become more evident through the pandemic. Therefore, radiologists should pay special attention to the clinical background and ensure active physician communication [33].

Consistent with the previous reports [34,35], we found that the patients with vaccination and booster dose had slighter clinical severity and shorter in-hospital time than unvaccinated cases in Delta and Omicron variants. This suggested that vaccination and booster shots are still effective despite the immune escape characteristics of the Delta and Omicron variants. Previous reports [8] showed that the vaccination was associated with a lower median CT severity score (CT-SS); unvaccinated patients had a median CT-SS of 11, while patients who had received a booster vaccine had a median CT-SS of five. However, we found no significant differences in pulmonary parenchymal involvement score regarding the vaccination status, similar to the study by Vincenza et al. [36]. Such a discrepancy in the results may be due to the individual differences in the samples included in different studies: most of our Omicron variant cases presented as a mild disease without pulmonary parenchymal involvement. Future multicenter, larger sample studies should be conducted to explore this point.

This study has the following limitations: Firstly, this is a single-center study with a small sample size. Second, some CT images of patients with original strain were of 5 mm thickness, which may have led to a CT score deviation and inaccurate lesion volume calculation by AI software. Thirdly, there are many variations in Omicron at present, so subgroup analysis was not conducted in the present study. Fourth, due to the retrospective nature of the study, some important information was not available, including BMI, smoking status, and cycle threshold values of the original strain. In addition, the vaccination time of participants could not be controlled, which may have an impact on the effect of the vaccine.

In conclusion, the present study indicated that the disease severity, extent of pneumonia, and typical appearance of pneumonia have gradually decreased from the original strain to the Delta variant and Omicron variant. Moreover, vaccination and booster vaccination still play a partial protective effect against variant strains, reducing the disease severity, lung involvement rate, and length of hospital stay.

## Figures and Tables

**Figure 1 biomedicines-11-00901-f001:**
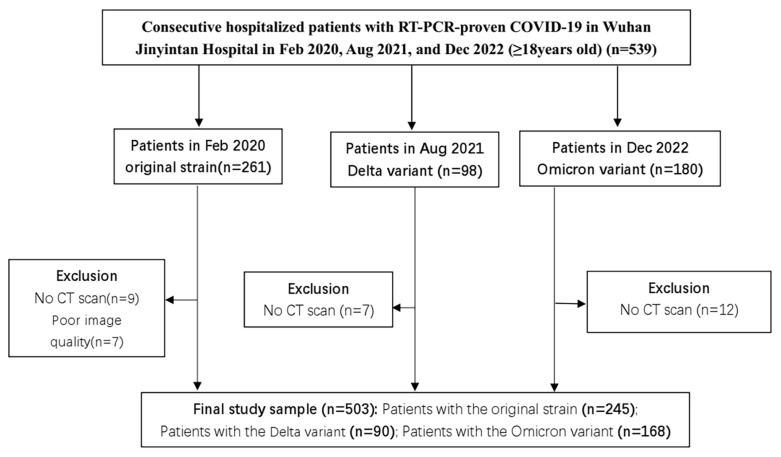
Flowchart of the patient’s inclusion and exclusion process.

**Figure 2 biomedicines-11-00901-f002:**
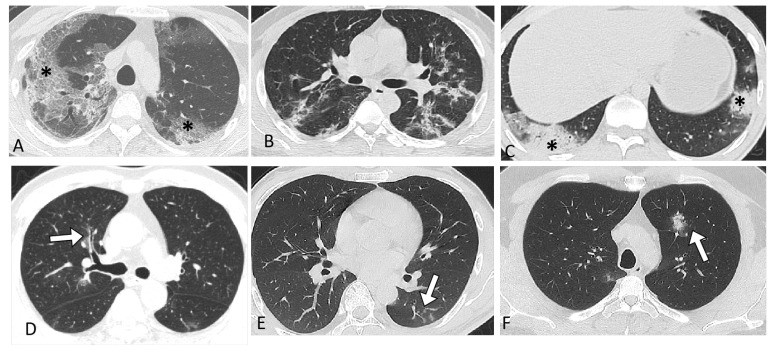
Axial chest CT scans in different patients with different SARS-CoV-2 strains infections: (**A**) The original strain of SARS-CoV-2 infection in a 38-year-old male with diffuse ground-glass shadows in both lungs with the thickened interlobular septum (crazy paving *) on day seven after symptom onset, mainly peripheral to both lungs; (**B**,**C**) Delta variant infection, (**B**) A 40-year-old male with multiple patchy solid shadows in both lungs, distributed along bronchial vascular bundles, on day five after symptom onset; (**C**) A 27-year-old male with multiple lumpy consolidations (*) in the subpleura of the lower lobes of both lungs on day three after symptom onset; (**D**–**F**) Omicron variant infection (**D**) A 54-year-old male with patchy GGOs in the upper lobe of the right lung with thickening of the surrounding bronchial wall on day four after symptom onset (arrow); (**E**) A 33-year-old male with a thin nodule in the dorsal segment of the left inferior lobe along the bronchus (tree bud sign, arrow); (**F**) 42, female, with scattered solid nodules in both lungs, with a few ground glass shadows around (halo sign, arrow).

**Table 1 biomedicines-11-00901-t001:** Comparison of demographic and clinical characteristics of different virus strains.

Clinical Characteristic	Original Strain (*n* = 245)	Delta Variant (*n* = 90)	Omicron Variant (*n* = 168)	*p* Value
Sex	134 (54.7%) ^a^	58 (64%)	102 (61%)	0.211
Age	58.1 ± 13.2 ^a,b^	39.1 ± 16.4 ^c^	45 ± 20	<0.001
Comorbidities	146 (59.6%) ^a,b^	18 (20%)	49 (29%)	<0.001
Diabetes	41 (16.8%) ^a,b^	1 (1.1%)	11 (6.6%)	<0.001
Hypertension	92 (37.7%) ^a,b^	7 (7.8%)	33 (20%)	<0.001
Cardiovascular and cerebrovascular disease	18 (7.4%) ^a,b^	0	17 (10%)	0.009
Chronic pulmonary disease	19 (7.8%) ^b^	2 (2.2%)	7 (4.2%)	0.094
Hepatopathy	9 (3.7%)	3 (3.3%)	6 (3.6%)	0.987
Nephropathy	10 (4.1%)	3 (3.3%)	4 (2.4%)	0.657
Received vaccination	--	68 (75.6%)	147 (88%)	0.014
Number of vaccination				<0.001
One dose	--	14 (15.6%)	6 (3.6%)	
Two doses	--	54 (60%)	33 (20%)	
Three doses	--	0	107 (64%)	
Clinical severity				<0.001
Mild type	10 (4.1%) ^a,b^	17 (19%)	95 (57%)	
moderate type	170 (69.4%)	64 (71%)	65 (39%)	
Severe or critical type	65 (26.5%) ^a,b^	9 (10%)	8 (4.8%)	
Clinical symptoms				
Fever	214 (88%)a,b	58 (64%)c	123 (75%)	<0.001
Maximum temperature	38 ± 1.6	37.4 ± 1.0c	38.2 ± 1.2	0.069
Cough	185 (75.5%) ^a,b^	41 (45.6%)	42 (25%)	<0.001
Sputum	79 (32.2%) ^a,b^	11 (12.2%)	34 (21%)	<0.001
Rhinorrhea	0 ^a,b^	4 (4.4%)	8 (4.9%)	0.006
Asthma	132 (53.9%) ^a,b^	5 (5.6%)	11 (6.5%)	<0.001
Throat pain	16 (6.6%) ^a,b^	21 (23.3%)	51 (31%)	<0.001
Diarrhoea	13 (5.3%)	10 (11.1%) ^c^	3 (1.8%)	0.006
Hyposmia	1 (0.4%) ^a^	5 (5.6%)	6 (3.6%)	0.008
Ophthalmia	0	3 (3.3%)	0	0.003
Weakness	73 (29.8%) ^a,b^	6 (6.7%)	52 (31%)	<0.001
Heart rate	94 ± 14 ^b^	87 ± 14	86 ± 12	<0.001
Respiratory rate	23 ± 5 ^a,b^	20 ± 1	21 ± 6	<0.001
Oxygen saturation	91 ± 10 ^a,b^	97.5 ± 1.3	98 ± 1.9	<0.001
SBP (mmHg)	132 ± 20 ^a,b^	126 ± 14	124 ± 12	<0.001
DBP (mmHg)	82 ± 11 ^b^	82 ± 11 ^c^	83 ± 9	0.009
Secondary infection	30 (12.2%) ^b^	5 (5.6%)	1 (0.6%)	
ICU	42 (17.1%) ^a,b^	2 (2.2%)	6 (3.6%)	<0.001
ARDS	65 (26.5%) ^a,b^	1 (1.1%)	0	<0.001
Hospital stay duration (days)	15 (11–23) ^a^	26.5 (19.6–34) ^c^	14 (10–17)	<0.001
Treatment				
Oxygen therapy	116 (47.3%) ^a,b^	10 (11.1%) ^c^	9 (5.4%)	<0.001
Endotracheal intubation	16 (6.6%) ^a,b^	0	0	0.001
Antiviral therapy	232 (94.7%) ^b^	86 (95.6%) ^c^	80 (48%)	<0.001
Antibiotic therapy	218 (89.0%) ^a,b^	7 (7.8%)	11 (6.5%)	<0.001
Hormone	163 (66.5%) ^a,b^	14 (15.7%) ^c^	3 (1.8%)	<0.001
Intravenous immunoglobulin	51 (20.9%) ^a,b^	2 (2.2%)	3 (1.8%)	<0.001
Mortality	35 (14.3%) ^a,b^	0	0	<0.001

The data are presented as frequency (percentage), medians (interquartile ranges), or median ± SD. ICU, intensive care unit; ARDS, acute respiratory distress syndrome. ^a^, *p* < 0.05 between the original strain and Delta strain; ^b^, *p* < 0.05 between original strain and Omicron strain; ^c^, *p* < 0.05 between Delta strain and Omicron strain.

**Table 2 biomedicines-11-00901-t002:** Comparison of CT scores and characteristics among different strains.

CT Characteristics	Original Strain (*n* = 245)	Delta Variant (*n* = 90)	Omicron Variant (*n* = 168)	*p* Value
CT scan thickness				<0.001
1.25 mm	140 (57.1%)	90 (100%)	168 (100%)	
5 mm	105 (42.8%) ^a^	0	0	
Abnormal CT	235 (95.9%) ^a,b^	73 (81%) ^c^	73 (44%)	<0.001
COVID-19 pneumonia imaging classification				<0.001
Typical appearance	228 (97%)	55 (75%)	48 (66%)	
Indeterminate appearance	3 (1.3%)	8 (11%)	13 (18%)	
Atypical appearance	4 (1.7%)	10 (14%)	12 (16%)	
Lesions involvement				<0.001
Unilateral	9 (3.8%) ^a^	17 (23%)	9 (12%)	
Bilateral	226 (96.2%) ^a^	56 (77%)	64 (88%)	
CT score of total lesions	14 (9.0–20.0) ^a,b^	6.0 (3.0, 8.5)	5.0 (3.0, 10)	<0.001
superior lobe of left lung	3.0 (2.0–4.0) ^a,b^	1.0 (0, 1.5) ^c^	1 (0, 1.0)	<0.001
inferior lobe of left lung	3.0 (2.0–4.0) ^a,b^	1.0 (0, 2.0)	1.0 (1.0, 2.0)	<0.001
superior lobe of right lung	3.0 (2.0–4.0) ^a,b^	0 (0, 1.0)	1.0 (1.0, 2.0)	<0.001
middle lobe of right lung	2.0 (1.0–3.5) ^a,b^	0 (0, 1.0)	0 (0, 1.0)	<0.001
inferior lobe of right lung	3.0 (2.0–4.0) ^a,b^	2 (1.0, 2.5)	2 (1.0, 2.5)	<0.001
CT score of GGO	11.0 (6.0–16.5) ^a,b^	4.0 (2.0, 7.0) ^c^	4 (1.5, 7.0)	
superior lobe of left lung	2.0 (1.0–3.0) ^a,b^	1.0 (0, 1.0)	1.0 (0, 2.0)	<0.001
inferior lobe of left lung	2.0 (1.0–4.0) ^a,b^	1.0 (0, 2.0)	1.0 (0, 2.0)	<0.001
superior lobe of right lung	2.0 (1.0–4.0) ^a,b^	0 (0, 1.0)	1 (0, 1.5)	<0.001
middle lobe of right lung	2.0 (1.0–3.0) ^a,b^	0 (0, 1.0)	0 (0, 1.0)	<0.001
inferior lobe of right lung	3.0 (2.0–4.0) ^a,b^	1.0 (1.0, 2.0) ^c^	1.0 (0, 2.0)	<0.001
CT score of consolidation	3.0 (0–8.5) ^a^	1.0 (0, 4.0) ^c^	2.0 (0, 5)	0.001
superior lobe of left lung	0 (0–2.0) ^a,b^	0 (0, 1.0)	0 (0, 0.25)	0.004
inferior lobe of left lung	1.0 (0–2.0) ^a,b^	0 (0, 1.0)	0 (0, 1.3)	<0.001
superior lobe of right lung	0 (0–2.0) ^a,^	0 (0, 0) ^c^	0 (0, 1.0)	0.003
middle lobe of right lung	0 (0–1.0) ^a,b^	0 (0, 0)	0 (0, 0.25)	<0.001
inferior lobe of right lung	1.0 (0–2.0) ^a^	0 (0, 1.0)	1.0 (0, 2.0)	0.008
CT score of linear opacities	3.0 (0–7.0) ^a,b^	0 (0, 2.0)	0 (0, 2.0)	<0.001
superior lobe of left lung	0 (0–1.0) ^a,b^	0 (0, 0)	0 (0, 0)	<0.001
inferior lobe of left lung	1.0 (0–2.0) ^a,b^	0 (0, 1.0)	0 (0, 1.0)	<0.001
superior lobe of right lung	0 (0–1.0) ^a,b^	0 (0, 0)	0 (0, 0)	<0.001
middle lobe of right lung	0 (0–1.0) ^a,b^	0 (0, 0)	0 (0, 0)	<0.001
inferior lobe of right lung	1.0 (0–2.0) ^a,b^	0 (0, 1.0)	0 (0, 1.0)	<0.001
Lung involvement				
superior lobe of bilateral lung	25 (10.2%) ^a,b^	16 (22%)	17 (23%)	0.007
inferior lobe of bilateral lung	220 (89.8%)	57 (78%)	56 (77%)	
Peripheral or subpleural	236 (96.3%) ^a,b^	62 (85%)	57 (78%)	
Degree of lesions involvement				<0.001
focal	9 (3.8%) ^a,b^	11 (15%)	19 (26%)	
Multiple	53 (22.6%)	34 (47%)	30 (41%)	
Diffused	173 (73.6%)	28 (38%)	24 (33%)	
Predominant CT pattern				0.093
GGO	176 (74.9%) ^a,b^	45 (62%)	48 (66%)	
Consolidation	44 (18.7%)	18 (25%)	20 (27%)	
Linear and reticulation	15 (6.4%)	10 (14%)	5 (6.8%)	
Crazy paving	79 (33.6%) ^a,b^	14 (19%)	10 (14%)	
Thickening of the bronchial wall	51 (21.8%)	7 (9.6%)	20 (28%)	0.009
Nodule or mass	8 (3.4%) ^a,b^	6 (8.2%)^c^	17 (23%)	<0.001
Tree-in-bud sign	0 ^a,b^	4 (5.5%)	6 (8.2%)	<0.001
Halo sign	4 (1.6%) ^a,b^	5 (6.8%)	10 (14%)	<0.001
pleural effusion	37 (15.7%) ^a^	8 (11%)	4 (5.5%)	0.063

The data are presented as frequency (percentage) or medians (interquartile ranges). GGO, ground-glass opacity; ^a^, *p* < 0.05 between original strain and Delta strain; ^b^, *p* < 0.05 between original strain and Omicron strain; ^c^, *p* < 0.05 between Delta strain and Omicron strain.

**Table 3 biomedicines-11-00901-t003:** Comparison of clinical and CT characteristics by different vaccinations status.

Virus Type	Delta Variant (*n* = 90)		*p* Value	Omicron Variant (*n* = 168)		*p* Value
Vaccination Status	Unvaccinated (*n* = 22)	Vaccinated (*n* = 68)		Less Than Three Doses (*n* = 61)	Three Doses (*n* = 107)	
Age	39.5 ± 22	38.8 ± 14.1	0.856	49 ± 24	42 ± 17	0.037
Sex			0.131			0.185
Male	12 (54.5%)	46 (68%)		33 (54%)	69 (65%)	
Female	10 (45.5%)	22 (32%)		28 (46%)	38 (36%)	
Comorbidities	7 (31.8%)	11 (16.2%)	0.111	24 (39%)	25 (23%)	0.028
Clinical type			0.04			0.015
Mild type	3 (13.6%)	19 (27.9%)		26 (43%)	69 (65%)	
moderate type	12 (54.5%)	37 (54.4%)		30 (49%)	35 (33%)	
Severe or critical type	5 (22.7%)	1 (1.5%)		5 (8.2%)	3 (2.8%)	
Fever	15 (68%)	43 (63%)	0.674	41 (67%)	63 (59%)	0.176
Maximum temperature	37.7 ± 1.1	37.2 ± 0.9	0.037	38 ± 1.3	38 ± 1.1	0.750
Hospital stay duration (days)	31 (24, 41)	25.5 (19, 32.8)	0.02	14 (10, 16)	13 (7, 18)	0.701
Abnormal CT	19 (86%)	54 (79%)	0.469	35 (57%)	38 (36%)	0.009
Lung involvement						
CT score of total lesions	5 (2, 10)	5.5 (3, 9.3)	0.891	5.5 (3, 9.3)	4 (3, 11)	0.673
CT score of GGOs	4.0 (1, 5.0)	4 (2, 7.0)	0.861	4 (2, 7)	3 (1, 8)	0.920
CT score of consolidation	1 (0, 3)	2 (0, 5.3)	0.229	2 (0, 5.3)	0 (0, 2)	0.825
CT score of linear opacities	0 (0, 2.0)	0 (0, 2)	0.723	0 (0, 2)	2 (0, 5)	0.274

The data are presented as frequency (percentage), medians (interquartile ranges), or mean ± SD. GGO, ground-glass opacity.

## Data Availability

The datasets used and/or analyzed during the current study are available from the corresponding author on reasonable request.

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
