# Peer review of "Comparative Analysis of Clinical and CT Findings in Patients with SARS-CoV-2 Original Strain, Delta and Omicron Variants"

_biomedicines, 2023, doi:10.3390/biomedicines11030901_

Round 1

Reviewer 1 Report

The manuscript “Comparative Analysis of Clinical and CT Findings in Patients with SARS-CoV-2 Alpha strain, Delta and Omicron Variants” presents a comprehensive comparison of the three prominent stains of COVID-19. It has a decent patient size and confirms the current common understanding about the three strains with the support of CT-scan and related experiments, which are widely accepted in scientific community. The scope and limitations of the study are well addressed.

Here are my comments/corrections/suggestions:

1.     In the objective section of the abstract, include “of COVID-19” after Alpha strain.

2.     The full description has to be first introduced before using acronyms.

3.     Origin and date of detection for alpha strain must also be given.

4.     Was delta variant first detected in October 2020 or December 2020?

5.     What does “1,2” indicate after November 24, 2021 in the introduction?

6.     Figure-1 should be embedded/referred in the text in section 2.1.

7.     Dates are wrong in Figure-1; February 2020 not 2022. December 2022, not 2021.

8.     It is not clear what the authors mean by “sex” in Table 1.

9.     It would be better if the authors include the BMI, smoking status, and ethnicity of the patients.

10.  Are cycle threshold values of alpha strain missing in the manuscript?

11.  In section 3.3, the sentence “Besides, the frequency…” is incomplete. Please complete it.

12.  In multiple places the sentence “and after adjustment for age, gender, underlying diseases” does not read well with the preceding sentence and should be grammar-checked.

13. In section 4, is “tropism” used as a noun or verb? Because that sentence is confusing and needs to be corrected. 

Author Response

Response to comments

Reviewer #1 (R1)

The manuscript "Comparative Analysis of Clinical and CT Findings in Patients with SARS-CoV-2 Alpha strain, Delta and Omicron Variants" presents a comprehensive comparison of the three prominent stains of COVID-19. It has a decent patient size and confirms the current common understanding about the three strains with the support of CT-scan and related experiments, which are widely accepted in the scientific community. The scope and limitations of the study are well addressed.

Here are my comments/corrections/suggestions:

R1-1.     In the objective section of the abstract, include “of COVID-19” after Alpha strain.

---   We sincerely thank the reviewer’s kind reminder. We have included “of COVID-19” accordingly.

R1-2.     The full description has to be first introduced before using acronyms.

---   We highly appreciate the reviewer’s suggestions. We have added the full description when first introduced.

R1-3.     Origin and date of detection for alpha strain must also be given.

---   We sincerely apologize for the confusion. The “alpha strain” has been revised as the "original strain”, since the COVID-19 patients in the current study came from the early stages of the global pandemic. The origin and date of detection for the original strain of COVID-19 had also been given in the introduction.

R1-4.     Was the delta variant first detected in October 2020 or December 2020?

---   We sincerely thank the reviewer’s kind reminder. We have checked the data when the delta variant was first detected and revised December 2020 as October 2020.

R1-5.     What does “1,2” indicate after November 24, 2021 in the introduction?

---   We sincerely apologize for the confusion. "1,2" was a typo and has been amended in the revised manuscript.

R1-6.     Figure-1 should be embedded/referred in the text in section 2.1.

---   Figure-1 has been embedded in the text in section 2.1.

R1-7.     Dates are wrong in Figure-1; February 2020 not 2022. December 2022, not 2021.

---   We sincerely thank the reviewer’s kind reminder. The dates have been corrected in Figure 1.

R1-8.     It is not clear what the authors mean by “sex” in Table 1.

---   We have changed it to "sex, man."

R1-9.     It would be better if the authors include the BMI, smoking status, and ethnicity of the patients.

---   We highly appreciate the reviewer’s suggestions. Since our study was a retrospective analysis, the BMI and smoking status information were unavailable. This limitation has been added to the revised manuscript.

R1-10.  Are cycle threshold values of alpha strain missing in the manuscript?

---   We highly appreciate the reviewer’s suggestions. Due to the retrospective analysis, the information of original strain's cycle threshold values was unavailable, which has been added in the limitation of the revised manuscript.

R1-11.  In section 3.3, the sentence “Besides, the frequency…” is incomplete. Please complete it.

---   We have completed the sentence as: “the frequency of indeterminate appearance(original strain,1.3% vs. Delta, 11% vs. Omicron,18%, p<0.001), and atypical CT appearance (original strain,1.7% vs. Delta, 14% vs. Omicron, 16%,p<0.001)were gradually increased.”

R1-12.  In multiple places, the sentence "and after adjustment for age, gender, underlying diseases" do not read well with the preceding sentence and should be grammar-checked.

---   We highly appreciate the reviewer’s suggestions. We have reorganized the statement as: “After controlling for gender, underlying diseases, and vaccination status with multiple logistics regression, SARS-CoV-2 strain type was significantly negatively correlated with clinical severity (OR=0.306, P < 0.001).”

R1-13. In section 4, is “tropism” used as a noun or verb? Because that sentence is confusing and needs to be corrected. 

---   We have completed the sentence as: “This may be related to the fact that omicron viruses tended to invade the upper respiratory tract over the lower respiratory tract.”

Reviewer 2 Report

This study aims to compare the clinical characteristics and chest CT findings of patients infected with the Omicron and Delta variants and the Alpha strain. It is an interesting topic, however the novelty in unclear. Also, the results shows that you are studying the impact of the booster vaccine on the clinical severity, but this was not stated in the abstract. Details regarding AI models should be added to the abstract. Also, more detailed about the proposed work is essential.

The abstract: I can see that the authors used artificial intelligence (AI) methods, however, they did not mention this in the abstract. Please state clearly what you did with AI and which AI methods were used. You stated that “The differences in general demographics, clinical severity, and 21 chest CT findings were analyzed” which techniques were used for the analysis.

Introduction

Is there any extra related works that could be added to the introduction?

Again, the authors seem to use AI but without mentioning it in the introduction and the impact of AI in COVID-19 diagnosis and differentiating between different variants of COVID-19. The authors as well could add related studies based on AI that were employed to distinguish different variants of COVID-19

Please add the paper organization by the end of the introduction.

Please highlight the novelty and contribution.

Methods and Materials

Please add more details regarding the subjects participating in the experiment and add samples to images of each class to the main manuscript.

Why did not the authors employ X-ray image?

This section lacks any details regarding the AI methods used in the study. It states that you are using U-Net with no further details. More details should be included.

Experimental Results

 I cannot see any discussion regarding the segmentation results like dice coefficient, sensitivity, and specificity.

Author Response

Reviewer #2 (R2)

This study aims to compare the clinical characteristics and chest CT findings of patients infected with the Omicron and Delta variants and the Alpha strain. It is an interesting topic, however the novelty is unclear. Also, the results show that you are studying the impact of the booster vaccine on the clinical severity, but this was not stated in the abstract. Details regarding AI models should be added to the abstract. Also, more detailed about the proposed work is essential.

R2-1.The abstract: I can see that the authors used artificial intelligence (AI) methods. However, they did not mention this in the abstract. Please state clearly what you did with AI and which AI methods were used. You stated that "The differences in general demographics, clinical severity, and 21 chest CT findings were analyzed" which techniques were used for the analysis.

---   We highly appreciate the reviewer’s suggestions. Some detail are not shown in the abstract due to the journal's word count requirement (<200 words). We have added the information on AI models and the booster vaccine on the clinical severity in the revised manuscript.

Introduction

R2-2 Is there any extra related works that could be added to the introduction?

---   We highly appreciate the reviewer’s constructive suggestions. We have added five related studies in the introduction of the revised manuscript.

R2-3 Again, the authors seem to use AI but without mentioning it in the introduction and the impact of AI in COVID-19 diagnosis and differentiating between different variants of COVID-19. The authors as well could add related studies based on AI that were employed to distinguish different variants of COVID-19.

 ---   We have added related studies based on AI in COVID-19 diagnosis in the introduction.

R2-4 Please add the paper organization by the end of the introduction.

---   We highly appreciate the reviewer’s comments. The paper organization has been added by the end of the introduction.

R2-5 Please highlight the novelty and contribution.

 ---   We highly appreciate the reviewer’s constructive suggestions. The novelty and contribution of the present study were emphasized in the resubmitted version.

Methods and Materials

R2-6  Please add more details regarding the subjects participating in the experiment and add samples to images of each class to the main manuscript.

--- We have added more details regarding the subjects participating in the experiment and added samples to images of each SARS-CoV-2 strain to the revised manuscript.

R2-7 Why did not the authors employ X-ray image?

---   We highly appreciate the reviewer’s comments. However, chest X-ray has been found to have a low sensitivity (69% [CI: 56-80%]) for early lung abnormities of COVID-19 [1], mainly used for diagnosis and classification of severe patients, and has been basically replaced by CT examination for COVID-19 in China.

[1] Wong HYF, Lam HYS, Fong AH-T, et al. Frequency and Distribution of Chest Radiographic Findings in COVID-19 Positive Patients[J]. Radiology, 0(0): 201160. doi: 10.1148/radiol.2020201160

R2-8 This section lacks any details regarding the AI methods used in the study. It states that you are using U-Net with no further details. More details should be included.

---   More details of AI methods were added in the method section.

Experimental Results

R2-9  I cannot see any discussion regarding the segmentation results like dice coefficient, sensitivity, and specificity.

--- InferRead MCT Lung, Infervision, Europe GmbH Wiesbaden Germany, an AI solution specifically developed for the diagnosis and management support of COVID-19 pneumonia. In a previous study [2], this AI algorithm achieved segmentation accuracy of a DICE coefficient of 0.8481 on internal test sets, which has been added to the revised manuscript. In our study, after the segmentation of lung abnormalities, two radiologists (XYH and LC, both with more than five years of experience in CT diagnosis) manually corrected the segmentation results for all patients.

[2] Huang L, Han R, Ai T, Yu P, Kang H, Tao Q, Xia L: Serial Quantitative Chest CT Assessment of COVID-19: A Deep Learning Approach. Radiology Cardiothoracic imaging 2020, 2(2):e200075.

Reviewer 3 Report

Excellent paper!

Minor corrections required:

- please only use capital letters for Alpha, Delta, and Omicron terms;

- minor grammar check in the introduction area.

Could you make some suggestions for the future regarding a possible  similar pandemic outbreak?    

Author Response

Reviewer #3 (R3)

Excellent paper!

Minor corrections required:

R3-1 Please only use capital letters for Alpha, Delta, and Omicron terms;

---   We highly appreciate the reviewer’s kind reminder. We have used capital letters for these terms throughout the manuscript.

R3-2 minor grammar check in the introduction area.

---   We highly appreciate the reviewer’s comments. We have checked the main text for grammar errors or typos in the resubmitted manuscript.

R3-3 Could you make some suggestions for the future regarding a possible similar pandemic outbreak?    

---   We highly appreciate the reviewer’s comments. Regarding a possible similar pandemic outbreak, humans need to strengthen preparedness for pandemic diseases, increase financing and investment for global pandemic preparedness, and improve surveillance and early warning systems.

Round 2

Reviewer 2 Report

The authors did not address some of my comments properly.

I asked to add a paragraph describing paper organization but the authors did not add it.

The authors did not add more details regarding the deep learning technique

Author Response

Response to comments

The authors did not address some of my comments properly.

---   We sincerely apologize for the previous negligence, and we have carefully revised the manuscript according to your comments.

R2-1 I asked to add a paragraph describing paper organization but the authors did not add it.

---   We sincerely apologize for not understanding your meaning before, we have added paper organization in the last paragraph of the introduction as following: “The rest of this paper was organized as follows. Section II showed how to include and exclude patients, the CT scanning method, and the statistical analysis used. We also compared the infection severity of patients with different vaccination statuses and quantified pneumonia by a deep-learning approach. In section III, the critical and statistically significant results are described. The significance of the results of this paper and relate the findings of this paper to previous research were discussed in Section IV. Finally, we conclude the paper in Section V.

R2-2 The authors did not add more details regarding the deep learning technique

---   We sincerely apologized for not specifying the deep learning technique, and we have added more details to the method section as following: “The core algorithm is based on a deep convolutional neural network structure and uses the U-net network structure as the core segmentation network [16]. UNet(https://lmb.informatik.uni-freiburg.de/people/ronneber/u-net), as a semantic segmentation framework, can effectively process the features of CT images of different scales and generate segmentation results. UNet + Xception structure is proposed to be adopted in the model structure. UNet is a semantic seg-mentation framework that can effectively process the features of CT im-ages of differ-ent scales and generate segmentation results. Xception is used as the backbone of UNet feature extraction. Xception can effectively avoid overfitting in the structure design. After the completion of training, the model can accurately segment the lung abnormalities area and calculate the percentage of pneumonia lesions in the overall lung volume.”